# Seeing our 3D world while only viewing contour-drawings

**Maddex Farshchi, Alexandra Kiba, Tadamasa Sawada** [ID] *

School of Psychology, National Research University Higher School of Economics, Moscow, Russia

* tsawada@hse.ru, tada.masa.sawada@gmail.com

## Abstract

Artists can represent a 3D object by using only contours in a 2D drawing. Prior studies have shown that people can use such drawings to perceive 3D shapes reliably, but it is not clear how useful this kind of contour information actually is in a real dynamical scene in which people interact with objects. To address this issue, we developed an Augmented Reality (AR) device that can show a participant a contour-drawing or a grayscale-image of a real dynamical scene in an immersive manner. We compared the performance of people in a variety of run-of-the-mill tasks with both contour-drawings and grayscale-images under natural viewing conditions in three behavioral experiments. The results of these experiments showed that the people could perform almost equally well with both types of images. This contour information may be sufficient to provide the basis for our visual system to obtain *much* of the 3D information needed for successful visuomotor interactions in our everyday life.

**Data Availability Statement:** All empirical data reported in this study are available from https://osf.io/t5jgb/.

**Funding:** This article was prepared within the framework of the Academic Fund Program at the National Research University Higher School of

## Introduction

Artists can represent a 3D scene with 3D objects by using only contours in a 2D contour-drawing and people can recognize the scene and objects reliably from such drawings [1–9]. There are computer vision algorithms that try to emulate this artists' skill and can generate contour-drawings from 2D photographic-images of 3D scenes [10, 11] and from 3D information contained in the scene [12, 13]. These contours represent an abrupt change of the luminance, color, or texture in the image and characteristic features in the 3D information. These characteristic features include self-occluding boundaries on the surface of objects [14–16], ridges on the surface [17, 18], as well as sharp edges on surfaces (see [12, 13] for reviews). Neither the luminance-polarity nor luminance-gradients are present in a contour-drawing.

Human beings can see the shape and position of a 3D object veridically when given only 2D drawings of it, and they can also recognize such objects reliably [1, 3–6, 19–27]. These well-known facts present a problem because according to Inverse-Problem Theory, the recovery of the shape of a 3D object from a 2D drawing is an ill-posed inverse problem. There are infinitely many 3D interpretations of a 2D contour-drawing. Note that line-drawings lack most depth cues, including binocular disparity, shading, and cast-shadows. This inverse problem can be resolved by imposing *a priori* constraints on the family of possible 3D interpretations [3, 5, 28–30]. Now, consider ordinary objects we see and use in our everyday life. Such objects are

Economics (HSE University) in 2019 (grant № 19-04-006, awarded to TS) and by the Russian Academic Excellence Project «5-100». https://www.hse.ru/science/scifund/nug/nug2019 The sponsors or funders play no role in the study design, data collection and analysis, decision to publish, or preparation of the manuscript.

**Competing interests:** The authors have declared that no competing interests exist.

not composed of a random scattering of points. They can be characterized by regularities of their shape, for example, their symmetry, volume, the planarity of their contours, and the presence of rectangular corners. These regularities, which introduce specific features into the drawing, also could be used to detect the presence and shape of an object [30–33]. Our visual system could make use of these regularities as the *a priori* constraints needed to recover the shape of a 3D object from a 2D contour-drawing of the object [34–38].

Prior studies that tested 3D perception from contour drawings have shown that people can obtain 3D information from the contour drawing, but it is not clear just how useful such contour information actually is in real dynamical 3D scenes in which ordinary objects are recognized and utilized under natural viewing conditions. These studies generated contour-drawings of objects taking care to avoid using degenerate views [5, 30], but note that in the real 3D scenes, objects will often be seen with degenerate views. Also note that people often change their viewing positions providing them with different views of the objects. Put simply, people can interact with the objects in real dynamical scenes.

Visual perception in real dynamical 3D scenes can be studied using the XR (Augmented-, Mixed-, and Virtual-Reality) technology. This XR technology can provide immersive experiences of a 3D scene that can be controlled by a computer. It has been shown that people can see the 3D information in the scene and they can interact within the scene on the basis of the visual information provided by the XR technology even if the scene is not fully photorealistic [39–41].

We developed an Augmented Reality (AR) device that can show a participant both a contour-drawing and a grayscale-image of a real dynamical 3D scene in an immersive manner (see [42–44] for earlier studies using AR devices to test the human visual system). The grayscale images were used as a control. They provided a baseline for the performance of a participant conducting our kind of tasks while wearing this device in our experiments. Our AR device allowed us to determine how well the participant can interact dynamically with objects in a scene, by using only contours in the contour-drawing or by using luminance-polarity and luminance-gradients in the gray-scale image.

## General methods

### AR device

The AR device used in this study showed, in an immersive manner, a contour-drawing and a grayscale image that represented a scene "out there" (Fig 1). The device was composed primarily of a smart phone (Lenovo Phab 2 Pro) and a wearable stereoscope (VR head-set). The phone ran on the Google Android OS which was equipped with an LCD screen and a camera located on the back of the screen. The two halves of the screen were seen individually by the two eyes of a participant who looked through the lenses of the stereoscope. The distance between the eyes and the screen, when the stereoscope was worn, was 8.5 cm. The screen's resolution was 1248 × 2560 pixels, and its size was 6.9 × 14.2 cm.

The phone's camera captured a photographic image of the scene in front of the participant. A contour-drawing and grayscale-image representing the scene were generated from this photographic image. This image was first converted to a grayscale-image $I_G$. Then, $I_G$ was passed through a set of image filters to generate the contour-drawing $I_C$:

$$I_C = 0.5|(I_G * B) * S_V| + 0.5|(I_G * B) * S_H|$$

where $B$ was a Gaussian filter and $Sv$ and $Sh$ were Sobel filters [10, 11, 45] that emphasized

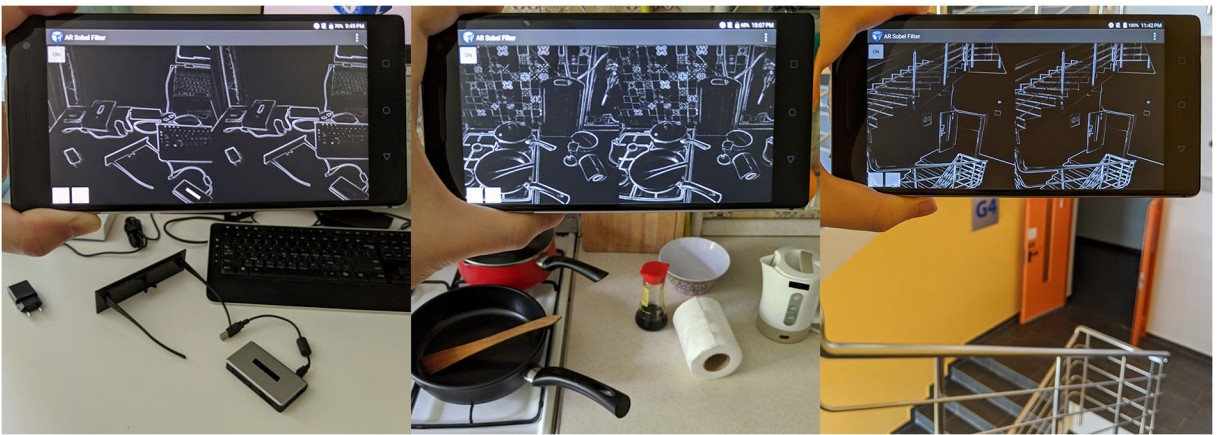

**Fig 1. Contour-drawings of real scenes generated by the AR device used in this study.** Copy-righted parts in the images have been blacked out.

vertical and horizontal edges:

$$B = \frac{1}{16} \begin{bmatrix} 1 & 2 & 1 \\ 2 & 4 & 2 \\ 1 & 2 & 1 \end{bmatrix}$$

$$S_V = \begin{bmatrix} -1 & 0 & +1 \\ -2 & 0 & +2 \\ -1 & 0 & +1 \end{bmatrix}$$

$$S_H = \begin{bmatrix} -1 & -2 & -1 \\ 0 & 0 & 0 \\ +1 & +2 & +1 \end{bmatrix}$$

Sobel filters were chosen because of their computational simplicity. This allowed our AR device to process the photographic images in near real-time. It is worth mentioning that an analogy of this algorithm with the visual system's process of edge detection in the primary visual cortex has been discussed [46]. This image process was implemented as an Android app using OpenCV library [47]. The resolution of the original photographic image and of the processed images was the same as the resolution of the screen ($1248 \times 2560$ pixels).

Two image segments ($1248 \times 1280$ pixels) taken from regions in the grayscale-image $I_G$ or in the contour-drawing $I_C$ were shown on the left and right halves of the screen (Fig 2). These regions were horizontal translations of one another in $I_G$ and $I_C$. The size of the translation $\Delta_S$ could be adjusted to allow the participant to fuse the retinal mages of the halves of the screen when the screen was viewed with the stereoscope. Note that the two halves of the screen were seen binocularly but binocular depth cues (binocular disparity and vergence) could not be used to perceive the 3D scene. These cues simply represented a frontoparallel plane but its effect on the immersive experience with the AR device seemed to be small [48].

A small wide-angle lens was attached to the camera to widen the camera's field of view. The visual angle of each image segment that was displayed on half of the screen was $53 \times 54°$ from the camera when this lens was attached. The visual angle of each screen half was $58 \times 59°$ from

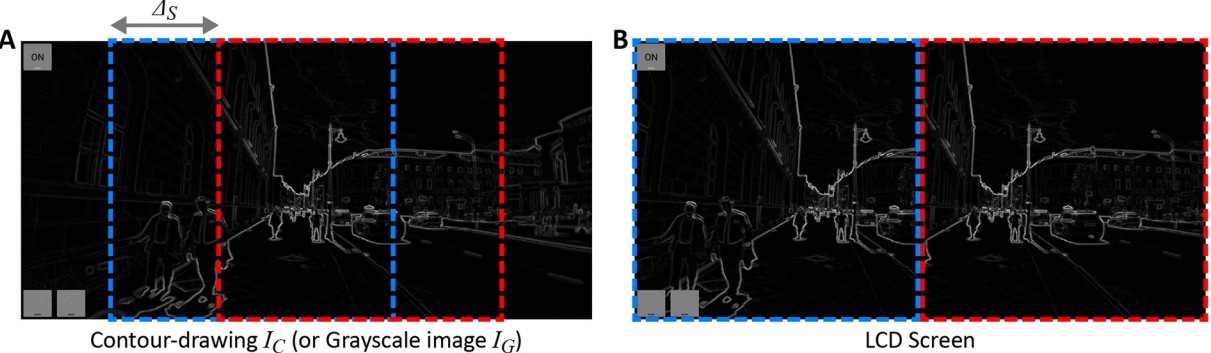

**Fig 2. (A) Regions of two image segments (blue and red) taken from the contour-drawing $I_C$, or from the grayscale-image $I_G$, and (B) an image composed of these segments on the LCD screen.** These regions are horizontal translations of one another for $\Delta_S$ in $I_G$ and $I_C$.

the eye of the participant. The refresh-rate of the screen was 10 Hz because of the time required to generate the grayscale-image and the contour-drawing. These processes also introduced a delay of 200–300 msec into the time required to refresh the screen. The refresh-rate (10 Hz) and the delay (200–300 msec) should be acceptable for an immersive experience when the AR device was used, but it could degrade a participants' interaction in the AR environment (see [49–51] for reviews; see the Appendix). Note that the contour-drawing was always generated regardless of whether the contour-drawing or the grayscale-image were shown on the screen. This made the refresh-rate and the delay constant with both types of images.

A shutter panel was also attached to the camera. This panel fully occluded the camera's view when it was closed. A clicking signal produced by pressing the middle button of a mouse was triggered when the shutter panel opened. This signal was used to signal the onset of a trial in the experiments.

## Procedure

The experiments were conducted in a well-lit room. The participant's tasks were different from one another, but all of the tasks required interacting with objects on a desk in front of the participant. During these tasks, the participant sat on a chair in front of this desk and viewed the objects on it through our AR device. Note that the participant could only see objects that were relevant to the task at hand during each trial. All other objects were hidden from view.

The two conditions with different image filters were blocked within each experiment and the participant put on the AR device with one of the image filters before each block. The block started with a training phase during which the participant was asked to look at their hands and to examine the scene through the device for 1 minute. The participant got used to viewing through the filter in this adaptation phase. All of the objects relevant to the tasks used in the experiments were hidden from the participants during the adaptation phase.

Before each trial, the shutter panel of the AR device was closed to occlude the participant's view (Fig 3). The trial began by opening the panel. The participant was asked to finish a given task as soon as possible and to press a large green button on a wall in front of the participant when s/he was finished. The response time between opening the shutter panel and pressing the green button was recorded.

Participants were 36 undergraduate students (aged 18 or over) in the Department of Psychology at the National Research University Higher School of Economics. All had normal or corrected-to-normal vision. There were 12 participants in Experiment 1, 12 participants in Experiment 2, and 12 participants in Experiment 3 (see https://osf.io/t5jgb/ for details). All

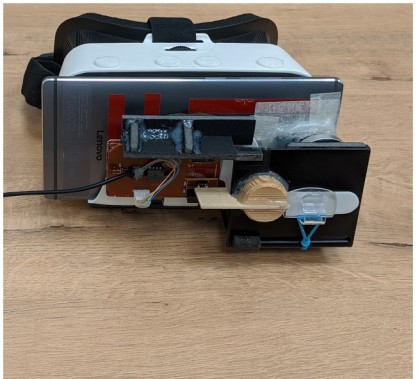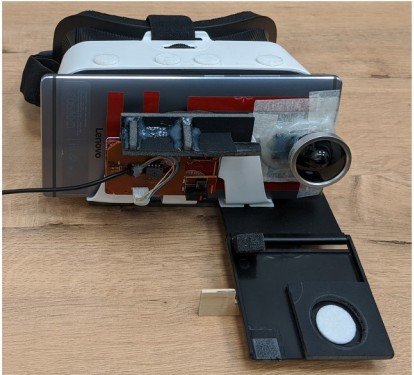

**Fig 3. The AR device used in this study with its shutter panel closed (left) and opened (right).**

were naïve with respect to the purpose of the study. Written informed consents were obtained from all of the participants. They were compensated with 100 Rubles for their participation.

The experiments described were conducted in accordance with the Code of Ethics of the World Medical Association (Declaration of Helsinki) and approved by the institutional review board (the HSE Committee on Interuniversity Surveys and Ethical Assess of Empirical Research).

## Experiment 1: Shape matching

In Experiment 1, we measured performance of a shape matching task with the image filters.

### Procedures

A participant was given 12 prism-shaped objects that were randomly oriented on a tray and a box with 12 holes whose shapes corresponded to the 12 individual objects (Fig 4). The shapes of all of the holes were different from one another. They were the same as the shapes of the cross-sections of the individual objects, and the objects could go through only their corresponding holes. The participant had to insert all of the objects into the box by finding their unique holes. This task required matching the shapes of the objects with the shapes of the holes. Note that the participant could not see any of these objects before the first trial. The nature of this task was explained to the participant by using an analogous toy before the experiment. Put simply, this task required recognizing the objects' shapes and the shapes of their holes and then coordinating their relative positions and orientations.

The experiment had 2 blocks consisting of 3 trials. There were 2 groups of 6 participants. The first group ran the block with the contour-drawing filter first. This was followed by the block with the grayscale-image. The second group ran the blocks in the opposite order. The participants were asked to rest for 5 minutes between the blocks during which they did not wear the AR device.

### Results

Fig 5 shows the averaged results observed in Experiment 1. The ordinate shows the response time. The abscissa shows the trials. The colors of the plots (blue and orange) represent the image filters (contour-drawing and grayscale image) and the styles of the plots represent the two groups of participants. The results were analyzed by using a three-way mixed-design ANOVA with repeated measures on two factors [52]: groups of participants, image filters, and

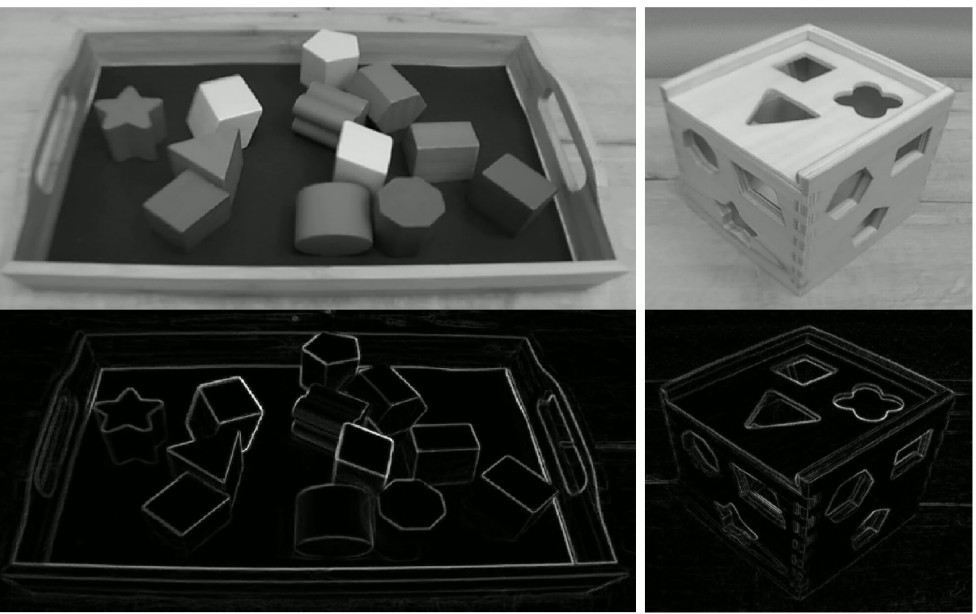

**Fig 4. Gray-scale images and contour-drawings of 12 prism-shaped objects on a tray (left) and the box with 12 holes (right) used in Experiment 1.** The shapes of the holes corresponded to the 12 individual objects.

trial numbers in each block (1, 2, and 3). The effect of the trial numbers ($F_{2,50} = 9.0$, $p = 0.00046 \times 7$, where 7 is multiplied for a Bonferroni correction, see [53]) and the interaction between the filters and groups ($F_{1,50} = 27$, $p = 3.5 \times 10^{-6} \times 7$) were significant. The results of the other effects were not significant: the filters ($F_{1,50} = 0.31$, $p = 0.58 \times 7$), the groups ($F_{1,10} = 0.016$, $p = 0.97 \times 7$), the trial numbers × filters ($F_{2,50} = 0.22$, $p = 0.80 \times 7$), the trial

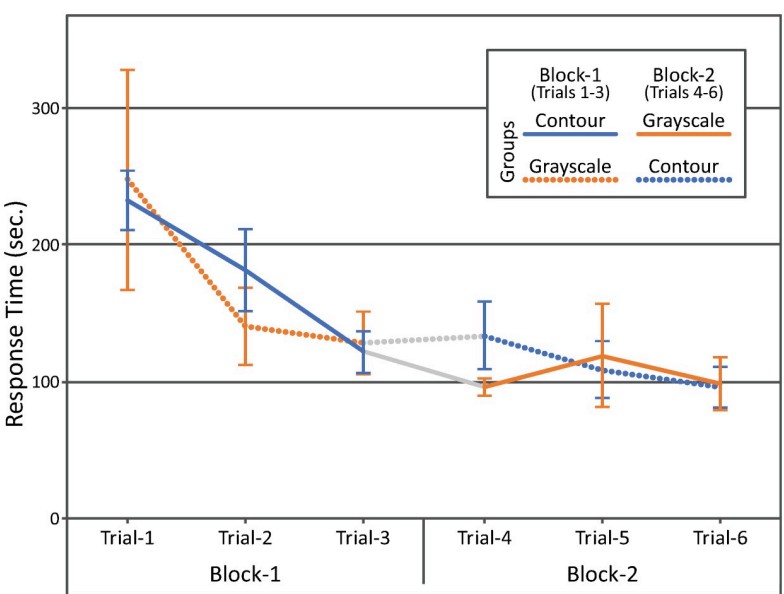

**Fig 5. The results obtained in Experiment 1.** The ordinate shows the response time and the abscissa shows the trials. The colors of the plots represent the image filters and the styles of the plots represent the groups of participants. The error bars show the standard errors across the participants. The 95 percent confidence intervals based on the *t*-distribution are 2.45 times of the standard errors ($CDF_t^{-1}(p = 0.975, n = 6) = 2.45$).

numbers × groups ($F_{2,50}$ = 1.4, $p$ = 0.26 × 7), the trial numbers × filters × groups ($F_{2,50}$ = 5.4, $p$ = 0.0075 × 7). (Note that $p$-values are adjusted to 1 if the $p$-values become larger than 1 after multiplying the Bonferroni factor.).

A posteriori test (Tukey) was performed to test the interaction between the filters and groups. The response time was shorter in the first block than in the second block: $p$ = 0.00091 for the group that ran the block with the contour-drawing filter first and $p$ = 0.0095 for the group that ran the block with the grayscale-image. The effect of the filter was significant neither in the first block ($p$ = 1.0) nor in the second block ($p$ = 1.0).

These results show that a human participant could conduct the shape matching task reliably with both a contour-drawing and a grayscale image that represented a scene "out there". We did not observe any difference in performance with both types of representation.

## Experiment 2: Object recognition

In Experiment 2, we used our image filters to measure performance in an object recognition task.

### Procedure

The participants sat in front of an open box and a collection of toys that were randomly oriented on a tray (Figs 6 and 7). These toys represented a variety of animals. The experimenter said the names of 3 target animals out loud, and the participant was asked to repeat these names while the shutter panel was kept closed to occlude the participant's view. When the panel was opened, the participant found the toys that represented 3 target animals, which had been chosen from the collection, and put them into the box. Put simply, this task only required that a participant could recognize an object s/he had not seen before.

This experiment had 2 blocks, each containing 4 trials. Four collections, consisting of a variable number of animals, were used (Figs 6 and 7). The order of the trials was determined by using the Latin-square method for each image filter. Ten animal toys were used in 3 of the 4 collections and 7 were used in the remaining collection. The target animals were chosen by using the following criteria: (i) no animal was used more than once throughout all blocks for each participant and (ii) individual animals in each collection were used at roughly equal frequencies throughout the experiment. (see https://osf.io/t5jgb/ for a list of the target animals used in the experiment).

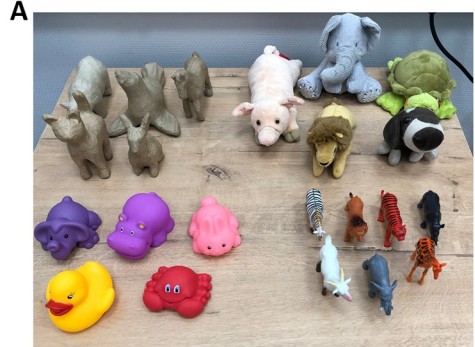
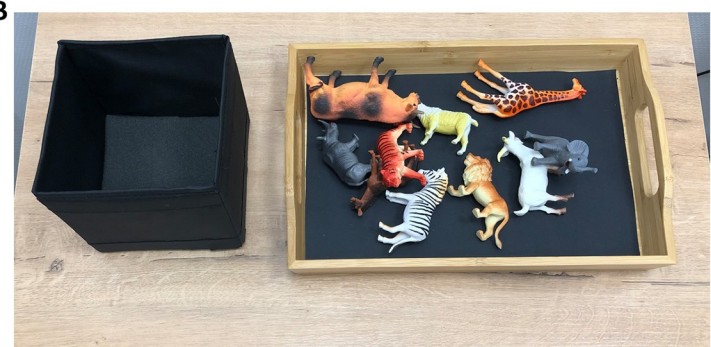

**Fig 6.** (A) 4 collections of animal toys used in Experiment 2: pulp-paper (left-top), stuffed (right-top), plastic-cartoon-like (left bottom), and plastic-realistic (right-bottom) animals. (B) The collection of toy animals on the tray, and the open box as they were arranged on the desk before each trial.

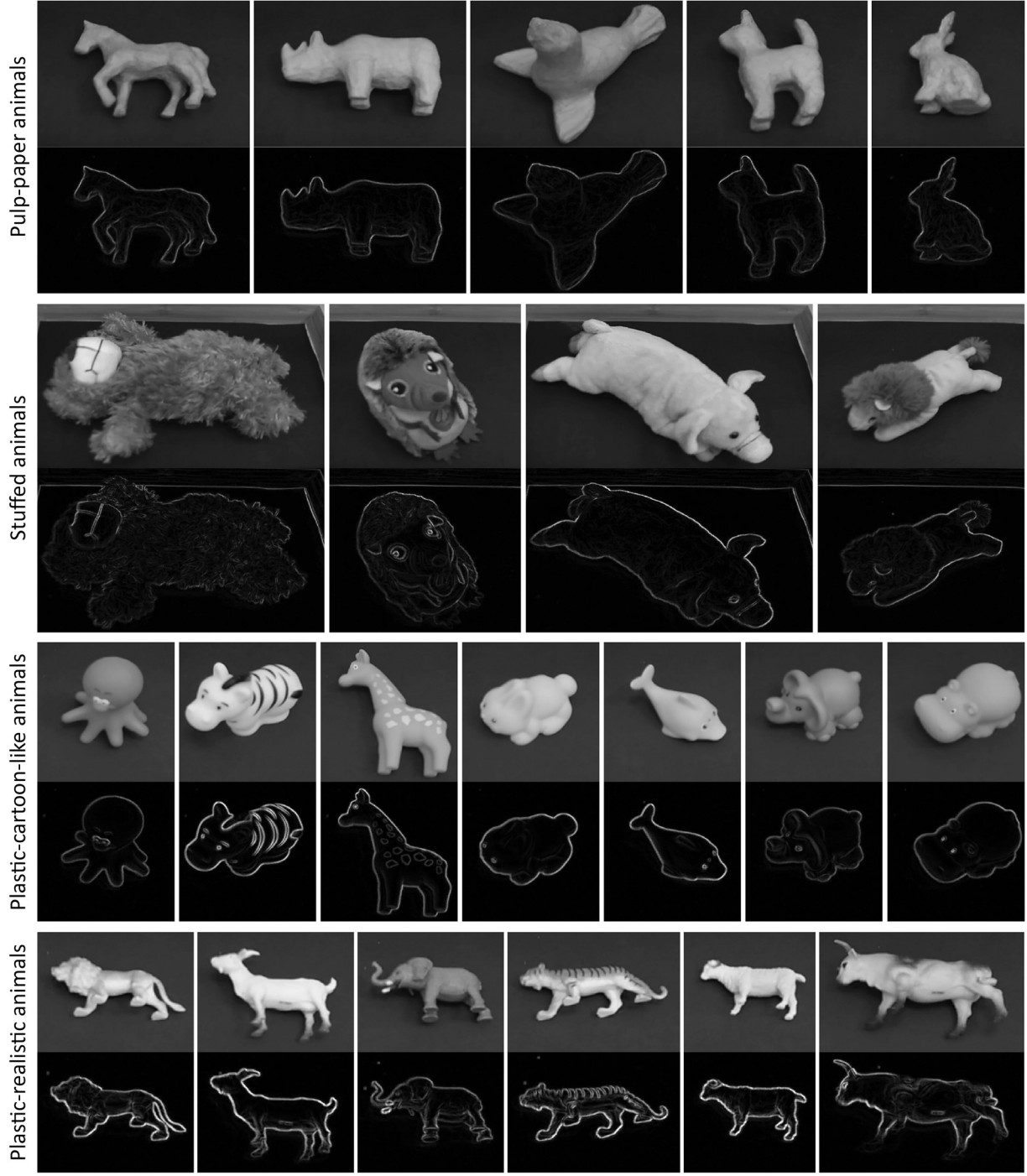

**Fig 7. Gray-scale images and contour-drawings of several samples taken from the 4 collections of animal toys used in Experiment 2.**

There were 2 groups of 6 participants. The first group ran the block with the contour-drawing filter first. This was followed by the block with the grayscale-images. The second group ran the blocks in the opposite order. The participants rested for 5 minutes between blocks. They did nor wear our AR device during these rest periods.

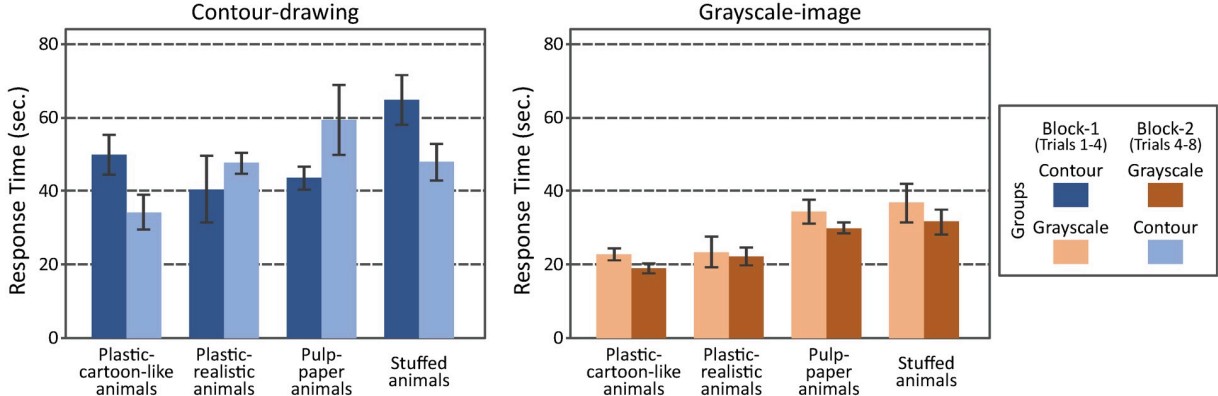

**Fig 8. The two panels of bar-graphs show the results obtained with both kinds of filters, *i.e.*, contour-drawing and grayscale-image.** The ordinate shows the response time. The abscissa shows the collections of toy animals. The brightness of the bars represents represent the two groups. The error bars show the standard errors across the participants. The 95 percent confidence intervals based on the *t*-distribution are 2.45 times of the standard errors ($CDF_t^{-1}(p = 0.975, n = 6) = 2.45$).

## Results

All participants recognized all of the objects. They made no errors in both image filter conditions, *i.e.*, contour-drawing and grayscale-image.

Fig 8 shows the averaged results observed in Experiment 2. The two panels of bar-graphs show the results obtained in both image filter conditions. The ordinate shows the response time. The abscissa shows the collections of animals. The brightness of the bars represents the two groups. The results were analyzed by using a three-way mixed-design ANOVA with repeated measures on two factors: groups of participants, image filters, and collections of toys. The two main factors were significant: the filters ($F_{1,70} = 75$, $p = 1.0 \times 10^{-12} \times 7$, where 7 is multiplied for a Bonferroni correction) and the toy collections ($F_{3,70} = 7.5$, $p = 0.00020 \times 7$). The results of the other effects were not significant: the groups ($F_{1,10} = 0.028$, $p = 0.87 \times 7$), the groups × toy collections ($F_{3,70} = 2.7$, $p = 0.051 \times 7$), the groups × filters ($F_{1,70} = 1.6$, $p = 0.21 \times 7$), the filters × toy collections ($F_{3,70} = 0.062$, $p = 0.98 \times 7$), and the groups × filters × toy collections ($F_{3,70} = 3.2$, $p = 0.030 \times 7$).

A posteriori test (Tukey) was performed to test the effect of the toy collections. The response time was shorter with the plastic-cartoon-like animals than with the stuffed animals ($p = 0.00075$) and with the pulp-paper animals ($p = 0.018$). The response time was shorter with the plastic-realistic animals than with the stuffed animals ($p = 0.0047$). The other pair-wise comparisons were not significant: plastic-cartoon-like vs. plastic-realistic ($p = 0.94$), plastic-realistic vs. pulp-paper ($p = 0.074$), paper-realistic vs. stuffed ($p = 0.74$).

## Experiment 3: Visuomotor coordination

In Experiment 3, we measured performance of a visuomotor interaction with objects in two tasks.

## Procedure

The participants performed two visuomotor interactions, one with tongs (Fig 9); the second with a brick (Fig 10). In the tongs task, the participant picked up and moved 7 objects from a tray to an open box. The tongs were used to minimize haptic information being used to

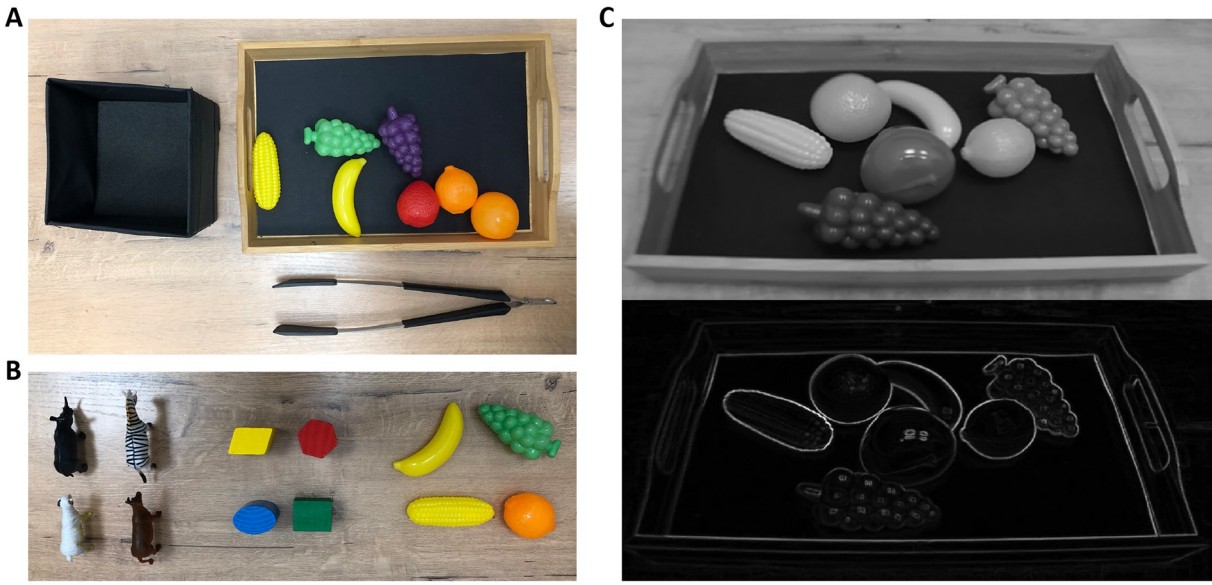

**Fig 9.** (A) A collection of 7 objects on a tray, and the open box as it was arranged on the desk before each trial of the tongs task in Experiment 3. (B) The 3 collections of objects used for the tongs task: plastic animals (left), wooden geometrical objects (middle), and plastic fruits-and-vegetables (right). (C) A gray-scale image and contour-drawing of the objects on the tray.

perform the task. The participants had to control the tongs, using only dynamical visual information provided by the tongs and by the objects on the tray. In the brick task, the participant was given an open box containing 12 rectangular bricks and used his/her hands to build a stack, with 4 layers of 3 bricks on the top of a stand.

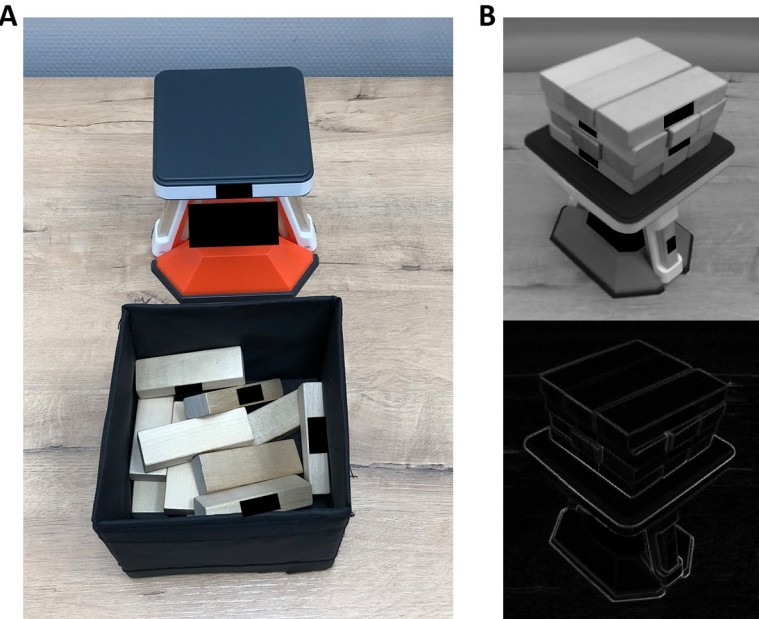

**Fig 10.** (A) The bricks and the stand as they were arranged on the desk before each trial of the brick task in Experiment 3. (B) A gray-scale image and contour-drawing of the 4 layers of 3 bricks built from the 12 bricks on the stand. Copy-righted parts in the images have been blacked out.

The experiment had 4 blocks: 2 image filters (contour-drawing and grayscale image) × 2 tasks (tongs and brick). Each block consisted of 3 trials. Three collections consisting 7 objects were used in the tongs task. These trials were repetitions of the blocks used in the brick task. The order of blocks and the order of trials within each block of the tongs task were randomized by using the Latin-square method with the following restriction: blocks with the contour-drawing and with the grayscale-image filters were conducted alternatively (see https://osf.io/t5jgb/). The participants rested for 5 minutes between blocks. They did not wear our AR device during these rest periods.

## Results

Fig 11 shows the averaged results for the tongs and brick tasks observed in Experiment 3. The ordinate shows the response time. The abscissa in Fig 11A shows the trials within each block. The abscissa in Fig 11B shows the collections of objects. The colors of the plots (blue and orange) represent the image filters (contour-drawing and grayscale image).

The results of the tongs task were analyzed by using a two-way between-subject-design ANOVA with repeated measures: image filters, and collections of objects. The main factor, object collections, was significant ($F_{2,55} = 49$, $p = 6.4 \times 10^{-13} \times 3$, where 3 is multiplied for a Bonferroni correction). The effect of the filters and the interaction between the filters and object collections were not significant: the filters ($F_{1,55} = 0.52$, $p = 0.48 \times 3$), the filters × object collections ($F_{2,55} = 0.18$, $p = 0.83 \times 3$). A posteriori test (Tukey) was performed to test the effect of the object collections. The response time was shorter with the plastic fruits-and-vegetables than it was with the plastic animals ($p = 1.2 \times 10^{-7}$) and with the wooden geometrical objects ($p = 4.1 \times 10^{-12}$). The response time was shorter with the plastic animals than with the wooden geometrical objects ($p = 0.0042$).

The results of the brick task were analyzed by using a two-way between-subject-design ANOVA with repeated measures: image filters, and trials. No effect was significant: the filters ($F_{1,55} = 0.0046$, $p = 0.95 \times 3$), the trials ($F_{2,55} = 3.2$, $p = 0.048 \times 3$), and their interaction ($F_{2,55} = 0.29$, $p = 0.75 \times 3$).

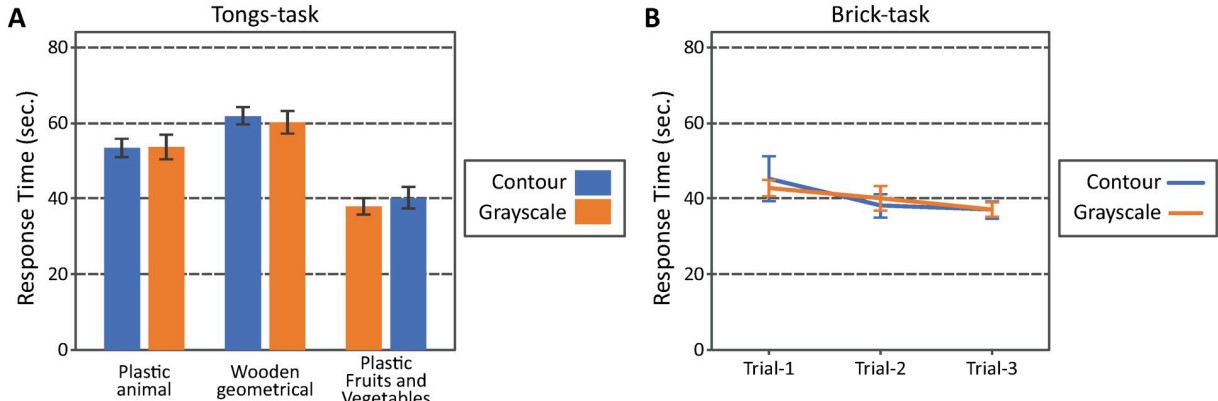

**Fig 11.** (A) The results of the tongs task obtained in Experiment 1. The ordinate shows the response time and the abscissa shows the collections of objects. The colors of the plots represent the image filters. The error bars show the standard errors across the participants. The 95 percent confidence intervals based on the *t*-distribution are 2.18 times of the standard errors ($CDF_t^{-1}(p = 0.975, n = 12) = 2.18$). (B) The results of the brick task obtained in Experiment 3. The ordinate shows the response time and the abscissa shows the trials. The colors of the plots represent the image filters. The error bars show the standard errors across the participants. The 95 percent confidence intervals based on the *t*-distribution are 2.18 times of the standard errors ($CDF_t^{-1}(p = 0.975, n = 12) = 2.18$).

## General discussion

We conducted three behavioral experiments that tested human performance when people interacted dynamically with objects in a real 3D scene, solely on the basis of a contour-drawing, or a grayscale-image that represented the scene. Contour-drawings were generated by applying an image filter to grayscale-images. The filter emphasized edges and eliminated luminance-gradients in the grayscale-image. The gray-scale images were used as a control. They provided a baseline for a participant's performance in our kind of tasks while wearing our AR device. The effect of the image filter was observed in the response time in an object recognition task (Experiment 2). Responses were slower with contour-drawings than with the grayscale-images, but note that participants could also recognize objects reliably from both contour-drawings and grayscale images. This difference in response time with these two types of images was not observed in our shape matching task (Experiment 1) or in our visuomotor coordination tasks (Experiment 3).

The tasks in Experiments 1 and 3 were designed so that the tasks required a dynamical visuomotor-coordination. The participant had to interact with multiple objects and to control their positions and orientations by using her/his hands. This kind of dynamical visuomotor-coordination is required in many run-of-the-mill tasks in which we use our hands in our everyday life. Note, however, that the tasks in this study did not require precise control of timing or quick reactions to unexpected events that could happen in a real-life scene. This kind of highly dynamical task could not be tested in this study because of technical limitations of our AR device. Note that such rapid processing of visual information is required in sports, and that it has been shown that the human visual system can process static contour-drawings very quickly [3, 6, 9]. So, it is possible that contour information is essential for visual processing in highly-dynamical tasks.

The results of Experiment 2 showed that a participant can recognize an object quite well when given a contour-drawing but recognition was even better when given its grayscale-image. This difference in performance can be attributed: (i) to the image filter used to generate the contour-drawing, and (ii) to the luminance-polarity and the luminance-gradient that are present in the grayscale-image but not in the contour-drawing. The image filter used a very simple algorithm that only emphasized the luminance edges on the basis of local information in the photographic image. This filter missed information in 2D images of a scene, such as edges between two isoluminant regions and between two regions that had different textures [54]. The filter could also miss contours that represent important features of the 3D information in a scene, for example, ridges on the surface of objects [17, 18]. This filter could also detect edges that are usually not drawn as contours in drawings made by an artist, for example, the edge of a shadow [55], and details of texture. These missed and redundant edges can degrade performance in an object recognition task [32, 56–59]. Note that the algorithm of the filter can be analogous to the visual system's process of edge detection in the primary visual cortex [46]. The human visual system must organize the edges in a retinal image in such a way that makes it possible for an observer to perceive the 3D information in the scene veridically [32, 58].

Also note that the luminance-polarity and the luminance-gradient present in the grayscale-image, but not in the contour-drawing, could also explain the difference in performance observed in Experiment 2. These two types of luminance information helped the visual system organize the luminance distributions and edges in the grayscale-images [42, 60–64]. Also, the luminance-gradient called "shading" could help the visual system to perceive the shape of the object's surface [18, 25, 27, 65–72] and to decompose the object on the basis of the surface's shape [73]. Note that this kind of luminance information did not improve performance in

Experiments 1 and 3, which suggests that the luminance-gradient and the luminance-polarity are not as important as the contours produced by luminance-edges in the retinal image are for perceiving 3D information in a real scene.

It is well-known that people can obtain 3D information from a contour drawing, but, until now, it was not clear how useful contour information actually is in a real dynamical scene. Our study shows that contour information, alone, is sufficient for ordinary people to perform a variety of run-of-the-mill tasks. We believe that our demonstration suggests that contour information, alone, may be sufficient to provide the basis for our visual system to obtain much of the 3D information needed for successful visuomotor interactions in our everyday life.

## Appendix

The AR device, used in this study, introduced some technical limitations in what the participants could do in Experiments 1, 2, and 3: specifically, the refresh-rate (10 Hz), the delay (200–300 msec), and the visual angles [49–51]. Furthermore, the Images shown on the screen of the device were achromatic, and binocular depth cues (binocular disparity and vergence) could not be used to perceive the 3D scenes. Also, the participants could move their heads less freely when they wore the AR device on their heads. All of these factors could degrade the participants' performance even in the baseline condition with the grayscale-images. Two of the authors (MF, TS) performed these tasks without wearing the AR device to get an idea about how difficult these tasks were under more natural conditions. Specifically, MF and TS ran sessions in Experiments 1, 2, and 3 without wearing the AR device. Their response times in these sessions are summarized in Table 1 (See Figs 5, 8 and 11 for comparison).

The response times of MF and TS were analogous to one another in the individual sessions and they were substantially shorter than the response times of the naïve participants even with the grayscale-image filter in Experiments 1, 2, and 3. It suggests that the participants'

**Table 1. Response times (sec) of MF and TS, under a more natural viewing condition, measured in sessions run when the AR device was not worn.** The response time with the two image filters in Experiments 1, 2, and 3 are also shown for comparison (average ± standard error).

| Exp. 1 | Trial-1 | Trial-2 | Trial-3 | Trial-4 | Trial-5 | Trial-6 |
|---|---|---|---|---|---|---|
| MF | 56.3 | 46.2 | 42.8 | 38.4 | 37.7 | 53.1 |
| TS | 56.0 | 39.8 | 39.1 | 40.1 | 37.5 | 36.0 |
| Contour | 232.4 ± 22.7 | 181.6 ± 30.7 | 121.7 ± 16.3 | 133.7 ± 25.6 | 108.7 ± 21.6 | 96.3 ± 15.8 |
| Grayscale | 248.9 ± 81.5 | 140.9 ± 29.1 | 128.8 ± 23.8 | 95.7 ± 7.5 | 118.8 ± 38.6 | 98.4 ± 19.8 |
| Exp. 2 | Plastic-cartoon-like animals | | Plastic-realistic animals | | Pulp-paper animals | Stuffed animals |
| MF | 6.7 | | 8.6 | | 7.6 | 7.5 |
| TS | 7.2 | | 5.9 | | 11.0 | 7.8 |
| Contour | 42.1 ± 4.4 | | 44.0 ± 4.8 | | 51.5 ± 5.5 | 56.4 ± 4.9 |
| Grayscale | 20.9 ± 1.3 | | 22.7 ± 2.5 | | 32.2 ± 2.0 | 34.2 ± 3.3 |
| Exp. 3: Tongs-task | Plastic animal | | Wooden geometrical objects | | Plastic fruits and vegetables | |
| MF | 12.5 | | 13.6 | | 12.6 | |
| TS | 13.0 | | 19.5 | | 14.8 | |
| Contour | 53.3 ± 2.7 | | 61.9 ± 2.4 | | 40.2 ± 3.1 | |
| Grayscale | 53.6 ± 3.5 | | 60.1 ± 3.3 | | 37.9 ± 2.4 | |
| Exp. 3: Brick-task | Trial-1 | | Trial-2 | | Trial-3 | |
| MF | 21.1 | | 16.5 | | 19.3 | |
| TS | 16.1 | | 16.3 | | 16.4 | |
| Contour | 45.2 ± 6.3 | | 38.1 ± 3.4 | | 37.1 ± 2.5 | |
| Grayscale | 42.8 ± 2.5 | | 40.1 ± 3.6 | | 37.1 ± 2.2 | |

performance with both types of image filters (contour-drawing and grayscale image) was suppressed by the loss of color information and as well as by technical factors in the AR device used in the experiments. These technical factors should be minimized to make the viewing condition of the experiments more natural. This will be addressed in a future study.

## Acknowledgments

We thank Svetlana V. Salomasova for helping to run the experiments reported in this study.

## Author Contributions

**Conceptualization:** Alexandra Kiba, Tadamasa Sawada.

**Data curation:** Maddex Farshchi.

**Formal analysis:** Maddex Farshchi, Tadamasa Sawada.

**Funding acquisition:** Tadamasa Sawada.

**Investigation:** Maddex Farshchi, Tadamasa Sawada.

**Methodology:** Maddex Farshchi, Alexandra Kiba, Tadamasa Sawada.

**Project administration:** Tadamasa Sawada.

**Resources:** Tadamasa Sawada.

**Software:** Alexandra Kiba.

**Supervision:** Tadamasa Sawada.

**Validation:** Tadamasa Sawada.

**Visualization:** Maddex Farshchi, Alexandra Kiba, Tadamasa Sawada.

**Writing – original draft:** Maddex Farshchi, Tadamasa Sawada.

**Writing – review & editing:** Maddex Farshchi, Alexandra Kiba, Tadamasa Sawada.

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
