## [Decision Letter · Decision Letter 0]

7 Aug 2020

PONE-D-20-19600

Seeing our 3D world while only viewing contour-drawings

PLOS ONE

Dear Dr. Tadamasa Sawada,

Thank you for submitting your manuscript to PLOS ONE. After careful consideration, we feel that it has merit but does not fully meet PLOS ONE’s publication criteria as it currently stands. Therefore, we invite you to submit a revised version of the manuscript that addresses the points raised during the review process.

We look forward to receiving your revised manuscript.

Kind regards,

Markus Lappe

Academic Editor

PLOS ONE

Journal Requirements:

2. We note that Figure(s) [10] in your submission contain copyrighted images. All PLOS content is published under the Creative Commons Attribution License (CC BY 4.0), which means that the manuscript, images, and Supporting Information files will be freely available online, and any third party is permitted to access, download, copy, distribute, and use these materials in any way, even commercially, with proper attribution. For more information, see our copyright guidelines: http://journals.plos.org/plosone/s/licenses-and-copyright.

1.    You may seek permission from the original copyright holder of Figure(s) [10] to publish the content specifically under the CC BY 4.0 license.

3. We note that Figure [3] includes an image of a [patient / participant / in the study]. 

4. Thank you for including your ethics statement:  "The experiments were conducted in accordance with the Code of

Ethics of the World Medical Association (Declaration of Helsinki) and approved by the institutional review board (IRB)"

Reviewers' comments:

Reviewer's Responses to Questions

**Comments to the Author**

1. Is the manuscript technically sound, and do the data support the conclusions?

Reviewer #1: Partly

Reviewer #2: Partly

2. Has the statistical analysis been performed appropriately and rigorously? 

Reviewer #1: Yes

Reviewer #2: Yes

3. Have the authors made all data underlying the findings in their manuscript fully available?

Reviewer #1: Yes

Reviewer #2: Yes

4. Is the manuscript presented in an intelligible fashion and written in standard English?

Reviewer #1: Yes

Reviewer #2: Yes

5. Review Comments to the Author

Reviewer #1: In this study participants performed simple visuomotor tasks in which they wore an AR device that showed them the visual field, but instead of capturing a full color photograph of the visual field the display showed a grayscale image or a the edges in the image. In each task, the participants’ behavior in the grayscale image and the edge image was not substantially different. The authors conclude that their study suggests that contour information is sufficient to the visual system to determine all 3D information that would be required for performing everyday tasks.

Overall, the experiments seem to be performed well, and the analyses were also performed correctly. It would be nice if the actual demographic information about participants was included in the text, and the readers were not just referred to the OSF repository.

The use of dynamic scenes seems to be an important point of the article, but I would guess that if the participant was shown only a static image, they would still be able to perform the task in experiment 2 and the tongs task of experiment 3, and that it would take equally as long to do the task in the line drawing condition as the grayscale image condition.

The claims in this paper rely on a negative finding. This makes it more difficult to justify the claims of the paper, especially with 12 participants per experiment. With a larger sample size the lack of a difference would be more persuasive. Also, to make the claim that this study shows that contours are sufficient to extract all necessary 3D information for every day tasks, the authors should spend some more time justifying that their three tasks generalize to other everyday tasks.

From previous work (M. Potter or I. Biederman, and from people communicating with drawings throughout history) we know that line drawings of static images are sufficient for image understanding. We also know that in a AR/VR environment people can successfully interact with their environment even if they are not photorealistic (Triesch, Ballard, Hayhoe, and Sullivan, 2003, many others too) – however I will note that this previous AR/VR work almost always gives the participant 3D information from binocular disparity. From our ability to understand cartoon videos, and the ability to even perceive intentionality and emotional content from simple line drawing videos (Hieder and Simmel) we know that people can understand dynamic line drawings. So I do not see anything unexpected about these results. I also am surprised that a study like this has not already been performed. So while I don’t see this as very novel or surprising, if it is has not been done by anyone else, then something like this should be published.

Reviewer #2: The paper is interesting, and the experiments are clearly described. The AR part is clear, even the stereoscopic part. Nevertheless, the authors must address the following concerns.

- rows 67-79 The authors implemented a simple algorithm for edge detection (not edges, but a combination of the image gradient), it would be very interesting to see how the results of their work might change as a function of the contour detection algorithms that can produce different kinds of contour (e.g. real edges, i.e. edges 1 pixel wide, black on white background or more human-like, i.e. edges more similar to the ones humans draw). The authors should discuss this point and try to do extend at least one of their experiments by using a different edge detection algorithm.

- rows 91-96 I wonder whether the low refresh rate and high lag has affected the results of the experiments, mainly since experiments are related to dynamical scene, when subjects interact with objects (i.e. the poor performance of the device has flattened the difference between the two conditions). This concern rises from my experience in AR/VR when the devices have poor performance. The authors should discuss (and take into consideration) this point.

- Experiment 1: Shape Matching It would be interesting to compare the subject performance in this AR experiment with respect to the baseline in real conditions (i.e. without wearing the AR device). This allow us both to have an idea of the reliability of the response time (e.g. it is so high that the difference between the condition are saturated) and to have an idea of the effect of the depth cue. The authors should discuss this point and try to do extend at least one of their experiments by comparing it with the result of one without wearing the AR device.

- rows 211-215: These results are affected by the kind of algorithms (conditions) the authors implemented, since the algorithm outputs depend on the object textures. I think this can not be completely related to the influence of contours on 3D interpretation of a scene.

- General discussion I am not totally convinced about the explanation of the authors, since their contours depend a lot on object textures, i.e. on the chosen algorithm, more than the effectiveness of the contours themselves (Experiment 2). Moreover, there is not a baseline without wearing the AR device. I think that the study could be more solid by following my previous suggestions.

6. PLOS authors have the option to publish the peer review history of their article (what does this mean?). If published, this will include your full peer review and any attached files.

Reviewer #1: No

Reviewer #2: No

---

## [Author Response · Author response to Decision Letter 0]

17 Sep 2020

Dear Dr. Markus Lappe,

We appreciate your handling our manuscript and two reviewers for reviewing the manuscript. All comments from the reviewers are addressed in this revision of the manuscript. We believe the manuscript is substantially improved thanks to the comments from the reviewers.

The suggestions are addressed point by point below with line numbers in the revised manuscript. All the revisions are in red with balloon comments in the revised manuscript “Revised Manuscript with Track Changes”.

E1. Style. Please ensure that your manuscript meets PLOS ONE's style requirements, including those for file naming. The PLOS ONE style templates can be found at

We formatted our manuscript according to PLOS ONE’s style requirements and confirmed that files are named properly.

E2. Figure 10. We note that Figure(s) [10] in your submission contain copyrighted images. All PLOS content is published under the Creative Commons Attribution License (CC BY 4.0), which means that the manuscript, images, and Supporting Information files will be freely available online, and any third party is permitted to access, download, copy, distribute, and use these materials in any way, even commercially, with proper attribution. For more information, see our copyright guidelines: http://journals.plos.org/plosone/s/licenses-and-copyright.

We blacked out all copyrighted parts in Figure 10 (L. 262).

E3. Figure 3. We note that Figure [3] includes an image of a [patient / participant / in the study]. 

As per the PLOS ONE policy (http://journals.plos.org/plosone/s/submission-guidelines#loc-human-subjects-research) on papers that include identifying, or potentially identifying, information, the individual(s) or parent(s)/guardian(s) must be informed of the terms of the PLOS open-access (CC-BY) license and provide specific permission for publication of these details under the terms of this license. 

We removed the photo with a person from Figure 3 (L. 129).

E4. Ethics statement. Thank you for including your ethics statement: "The experiments were conducted in accordance with the Code of Ethics of the World Medical Association (Declaration of Helsinki) and approved by the institutional review board (IRB)"

We revised the ethics statement to include the full name of the board (the HSE Committee on Interuniversity Surveys and Ethical Assess of Empirical Research). The revised statement was copied to the “Ethics Statement” field of the submission form (L. 147).

E5. Ethics statement. Please provide additional details regarding participant consent. In the Methods section, please ensure that you have specified (1) whether consent was informed and (2) what type you obtained (for instance, written or verbal). If your study included minors, state whether you obtained consent from parents or guardians. If the need for consent was waived by the ethics committee, please include this information.

We revised the ethics statement to specify that written informed consent was obtained from all the participants (L 146). All participants were undergraduate students (aged 18 or over) (L. 143).

E6. Figures 1, 2, 4, 6, 7, 9 and 10. Copyright

We confirm that we produced all the figures in this manuscript specifically for this manuscript. We also blacked out all potentially copyrighted parts in Figure 1 (L. 81).

Reviewers' comments:

Reviewer #1:

Thank you very much for reviewing our manuscript and for the constructive feedback. We were especially happy to receive your positive evaluation of our study. We also thank you for sharing reference information about prior relevant studies. We revised our manuscript, taking all of your suggestions into account point by point (see below).

In this study participants performed simple visuomotor tasks in which they wore an AR device that showed them the visual field, but instead of capturing a full color photograph of the visual field the display showed a grayscale image or the edges in the image. In each task, the participants’ behavior in the grayscale image and the edge image was not substantially different. The authors conclude that their study suggests that contour information is sufficient to the visual system to determine all 3D information that would be required for performing everyday tasks.

Overall, the experiments seem to be performed well, and the analyses were also performed correctly.

R1a. It would be nice if the actual demographic information about participants was included in the text, and the readers were not just referred to the OSF repository.

Text explaining the demographic information of our participants was added (L. 140). The participants were 36 undergraduate students in the Department of Psychology at the National Research University Higher School of Economics. All had normal or corrected-to-normal vision, and all were naïve with respect to the purpose of the study. No other personal information was collected.

R1b. The use of dynamic scenes seems to be an important point of the article, but I would guess that if the participant was shown only a static image, they would still be able to perform the task in experiment 2 and the tongs task of experiment 3, and that it would take equally as long to do the task in the line drawing condition as the grayscale image condition.

Thank you for raising this issue. The tongs task in Experiment 3 was designed to make haptic information useless when the task was performed. The participants used the tongs, so only dynamical visual information provided by the tongs and the objects on the tray was available. The Procedure section in Experiment 3 was revised to clarify this point (L. 245). 

We think that the object recognition task in Experiment 2 could be performed to some extent on the basis of static views of objects. But, in real dynamical scenes, people can interact with objects and they can change their view of the objects if they could not recognize the objects from their original view. A paragraph was added in the Introduction discussing object recognition in real dynamical scenes (L. 54, see also Comment R1e).

R1c. The claims in this paper rely on a negative finding. This makes it more difficult to justify the claims of the paper, especially with 12 participants per experiment. With a larger sample size the lack of a difference would be more persuasive.

We agree with the reviewer. We revised the Abstract (L.21) and Discussion (L. 290, 336) to address this concern.

R1d. Also, to make the claim that this study shows that contours are sufficient to extract all necessary 3D information for every day tasks, the authors should spend some more time justifying that their three tasks generalize to other everyday tasks.

We added a paragraph discussing the generalization of the tasks as well as its limitations in the Discussion (L. 299).

R1e. From previous work (M. Potter or I. Biederman, and from people communicating with drawings throughout history) we know that line drawings of static images are sufficient for image understanding. 

Prior studies (including studies by M. Potter and I. Biederman) that tested 3D perception from contour drawings used clean contour-drawings of objects taking care to avoid using degenerate views of the objects. Now note that in the real 3D scenes, objects will often be seen with degenerate views (see Comment R1b). Also, contour-drawings that are automatically generated from photographic images of a real scene often lack important contours and have redundant contours. We revised the text in the Introduction (L. 54) and Discussion (L. 312) that discusses the differences between contour-drawings made by artists for human observers and contour-drawings generated by computer algorithms.

Thank you for the references to these prior studies. We added your references to prior studies about the visual perception of contour-drawings including the studies by M. Potter and of I. Biederman. Note that Biederman (1987) was cited in our original submission.

R1f. We also know that in a AR/VR environment people can successfully interact with their environment even if they are not photorealistic (Triesch, Ballard, Hayhoe, and Sullivan, 2003, many others too) – however I will note that this previous AR/VR work almost always gives the participant 3D information from binocular disparity.

We added a paragraph about the nature of an immersive experience of a 3D scene with XR technology and how people interact with such scenes the basis of visual information provided by this technology (L. 61). We also added references to prior studies (including Triesch, Ballard, Hayhoe, & Sullivan, 2003) that discussed using the XR technology to study the visual perception.

It is critical to control the visual stimuli used systematically when the human visual system is studied. If someone wants to study the effect of degrading the photorealism of stimuli on the perception of the stimuli, the way the photorealism is degraded must be systematically controlled. We did it by using two types of image filters. We also examined the effect of the degrading by comparing the participants’ performance with both types of filters. We revised the text to make this point clear (L. 21, 116, 290, 344).

R1g. From our ability to understand cartoon videos, and the ability to even perceive intentionality and emotional content from simple line drawing videos (Hieder and Simmel) we know that people can understand dynamic line drawings. 

This comment is related to two unsolved questions: (i) how well can artists represent 3D scenes and 3D objects by using only contours in their drawings and (ii) how well can an artists’ skill be emulated by computer algorithms. Generating good contour-drawings from a 2D image of a 3D scene or acquiring 3D information of the scene is a non-trivial task in Computer vision. Our study examined how well people can perceive the 3D information contained in a real scene when a contour-drawing of it was automatically generated by a simple computer algorithm. An analogy of this algorithm with the visual system's process of edge detection in the primary visual cortex has been discussed in some prior studies (L. 99). We revised the text in the Introduction (L. 33) and Discussion (L. 316) to explain the difference between an artists’ drawing of a contour and a contour-drawing made by a computer algorithm.

R1h. So I do not see anything unexpected about these results. I also am surprised that a study like this has not already been performed. So while I don’t see this as very novel or surprising, if it is has not been done by anyone else, then something like this should be published.

We hope that our revision of the manuscript addressed all of your concerns and made this study more interesting. We appreciate the comments and suggestions that you made.

Reviewer #2:

Thank you very much for reviewing our manuscript and for your constructive feedback. We were pleased to see your interest to our study. We revised our manuscript to tale all of your suggestions into account point by point (see below).

The paper is interesting, and the experiments are clearly described. The AR part is clear, even the stereoscopic part. Nevertheless, the authors must address the following concerns.

R2a. rows 67-79 The authors implemented a simple algorithm for edge detection (not edges, but a combination of the image gradient), it would be very interesting to see how the results of their work might change as a function of the contour detection algorithms that can produce different kinds of contour (e.g. real edges, i.e. edges 1 pixel wide, black on white background or more human-like, i.e. edges more similar to the ones humans draw). The authors should discuss this point and try to do extend at least one of their experiments by using a different edge detection algorithm.

Thank you very much for this suggestion. We tried some other filters for edge detection, i.e., canny, but our AR device could not process these filters as quickly as a Sobel filter. The Sobel filter, introduced in 1968, is one of the simplest algorithms that can be used to emphasize edges in an image. The simplicity of the filter allowed our AR device to process photographic images from a camera in near real-time. We added text explaining the reason we chose the Sobel filter (L. 95).

Detecting important edges, while removing redundant edges, from a photographic image and generating a good contour-drawing from the image, or from a 3D model of a scene, are on-going research topics in Computer vision. We could say that the newer algorithms are better. We revised the text in the Introduction (L. 33) and the Discussion (L. 316) that address this issue.

R2b. rows 91-96 I wonder whether the low refresh rate and high lag has affected the results of the experiments, mainly since experiments are related to dynamical scene, when subjects interact with objects (i.e. the poor performance of the device has flattened the difference between the two conditions). This concern rises from my experience in AR/VR when the devices have poor performance. The authors should discuss (and take into consideration) this point.

We understand this reviewer’s concern. Based on the results of prior studies on the human factors that arise when XR technology is used, such as the refresh rate and the lag of our AR device, that despite the fact that it was acceptable, these factors could affect our subject's immersive experience. We revised the text in the General methods to address this concern (L. 114). We also discussed it in a new appendix (L. 338, see Comment R2c).

Note that the contour-drawing was always generated regardless of whether the contour-drawing or the grayscale-image were shown on the screen. This control made the refresh rate and delay consistent across the conditions of image filters. We added text explaining this control in the General methods (L. 116).

R2c. Experiment 1: Shape Matching It would be interesting to compare the subject performance in this AR experiment with respect to the baseline in real conditions (i.e. without wearing the AR device). This allow us both to have an idea of the reliability of the response time (e.g. it is so high that the difference between the condition are saturated) and to have an idea of the effect of the depth cue. The authors should discuss this point and try to do extend at least one of their experiments by comparing it with the result of one without wearing the AR device.

Two of the authors (MF, TS) ran sessions in Experiments 1, 2, and 3 without wearing the AR device to get an idea about how difficult these tasks were under more natural conditions. We added the new appendix section reporting these sessions (L. 338). Note that it is very difficult to test any naïve participants in such an interactive experiment in the current pandemic situation. 

R2d. rows 211-215: These results are affected by the kind of algorithms (conditions) the authors implemented, since the algorithm outputs depend on the object textures. I think this cannot be completely related to the influence of contours on 3D interpretation of a scene.

We expanded our discussion in the General discussion (L. 316) about the differences between contour-drawings drawn by artists and generated by computer algorithms (see also Comment R2a, R1e, R1g). 

R2e. General discussion I am not totally convinced about the explanation of the authors, since their contours depend a lot on object textures, i.e. on the chosen algorithm, more than the effectiveness of the contours themselves (Experiment 2). Moreover, there is not a baseline without wearing the AR device. I think that the study could be more solid by following my previous suggestions.

This study addresses the difference in performance observed with a contour-drawing and with a grayscale-image. The gray-scale images were used as a control. They provided a baseline for a participant's performance in our kind of tasks while wearing our AR device. We revised the text in the Abstract (L. 21) and the General discussion (L. 290) to make this point clearer.

Two of the authors (MF, TS) ran sessions in Experiments 1, 2, and 3 without wearing the AR device and these sessions are reported in the new appendix section (L. 338, see Comment R2c).

We believe your concerns about the quality of contour-drawings used in this study are addressed well in our replies to Comments R2a and R2d (see also Comments R1e and R1g).

We hope that our revision of the manuscript addressed all of your concerns. We appreciate the comments and suggestions that you made.

---

## [Decision Letter · Decision Letter 1]

19 Oct 2020

PONE-D-20-19600R1

Seeing our 3D world while only viewing contour-drawings

PLOS ONE

Dear Tadamasa,

I am happy to report that both reviewers are essentially satisfied with your revision. Reviewer 2 has a few minor points that you should be able to address easily. I will accept the paper once these minor changes have been made.

We look forward to receiving your revised manuscript.

Best regards,

Markus

---

Markus Lappe

Academic Editor

PLOS ONE

Reviewers' comments:

Reviewer's Responses to Questions

**Comments to the Author**

1. If the authors have adequately addressed your comments raised in a previous round of review and you feel that this manuscript is now acceptable for publication, you may indicate that here to bypass the “Comments to the Author” section, enter your conflict of interest statement in the “Confidential to Editor” section, and submit your "Accept" recommendation.

Reviewer #1: All comments have been addressed

Reviewer #2: (No Response)

2. Is the manuscript technically sound, and do the data support the conclusions?

Reviewer #1: Yes

Reviewer #2: Yes

3. Has the statistical analysis been performed appropriately and rigorously? 

Reviewer #1: Yes

Reviewer #2: Yes

4. Have the authors made all data underlying the findings in their manuscript fully available?

Reviewer #1: Yes

Reviewer #2: Yes

5. Is the manuscript presented in an intelligible fashion and written in standard English?

Reviewer #1: Yes

Reviewer #2: Yes

6. Review Comments to the Author

Reviewer #1: The authors addressed all of our concerns. They added relevant literature, demographic information, and clarifications about the study. As the study relies on a null finding, they slightly toned down their claims to a more appropriate level. The study may have benefited from a Bayesian analysis to support the claims more strongly. Overall, I see no technical problems with the paper as it is now.

Reviewer #2: The authors addressed the concerns I raised in my review in a satisfactory way (by looking at the answers to the other reviewer too).

In particular, I pointed out that it should interesting to compare the subject performance in these AR experiments with respect to the baseline in real conditions (i.e. without wearing the AR device). The authors replied that it is very difficult to test any naïve participants in such an interactive experiment in the current pandemic situation. They added an Appendix where two of the authors (MF, TS) ran sessions in Experiments 1, 2, and 3 without wearing. In normal situation this is not acceptable, but in the current situation I think this is an added value for the paper. Moreover, the important issue related to the use of only one edge detector, which can be solved by running the experiments by using a different algorithm, is hampered by the pandemic situation. Thus, it is fine for me again.

However, in the Appendix the authors should add a row to each table in order to add the average performances of the experiments with the AR device to simplify the comparison (I think that the “See Figs. 5, 8, 11 for comparison” is not enough). Moreover, they should add a short comment about the comparison. I think that this point (at least) is important to improve the paper.

Then, the work can be published for me.

7. PLOS authors have the option to publish the peer review history of their article (what does this mean?). If published, this will include your full peer review and any attached files.

Reviewer #1: No

Reviewer #2: No

---

## [Author Response · Author response to Decision Letter 1]

31 Oct 2020

Dear colleagues,

We appreciate you for reviewing our manuscript and suggestions you made. We were delighted to see that both of the reviewers are mostly satisfied with our last revision. We also thank the reviewers for understanding our situation. All the suggestions are addressed point by point below with line numbers in the revised manuscript. All the revisions are in red with balloon comments in the revised manuscript “Revised Manuscript with Track Changes”.

Reviewer #1:

R1a. The study may have benefited from a Bayesian analysis to support the claims more strongly.

Thank you very much for this suggestion. Note that the results of the experiments reported in this study were analyzed by using multi-way repeated-measure ANOVA but Bayesian alternatives of multi-way repeated-measure ANOVA is still under discussion (see Nathoo & Masson, 2016 for discussion). Instead, we added information of confidence intervals based on the t-distribution in the manuscript (l. 186-187, l. 238-240, l. 290-296; see Francis, 2017 for comparison between the conventional t-test and its Bayes alternative). We believe it provides better quantitative information about our results.

Reviewer #2:

R2a. However, in the Appendix the authors should add a row to each table in order to add the average performances of the experiments with the AR device to simplify the comparison (I think that the “See Figs. 5, 8, 11 for comparison” is not enough). Moreover, they should add a short comment about the comparison. I think that this point (at least) is important to improve the paper.

Thank you very much for this suggestion. We revised Table 1 (l. 366-371) to show average performance under the contour-drawing and grayscale-image conditions in the main experiments.

We also added a short paragraph discussing the difference between Appendix and the experiments reported in the main text of the manuscript.

References

Francis, G. (2017). Equivalent statistics and data interpretation. Behavior research methods, 49(4), 1524-1538.

Nathoo, F. S., & Masson, M. E. (2016). Bayesian alternatives to null-hypothesis significance testing for repeated-measures designs. Journal of Mathematical Psychology, 72, 144-157.

---

## [Editor Report · Decision Letter 2]

5 Nov 2020

Seeing our 3D world while only viewing contour-drawings

PONE-D-20-19600R2

Dear Tadamasa,

I am pleased to inform you that your manuscript has been judged scientifically suitable for publication and will be formally accepted for publication once it meets all outstanding technical requirements.

With best wishes,

Markus

Markus Lappe

Academic Editor

PLOS ONE
---

## [Editor Report · Acceptance letter]

9 Nov 2020

PONE-D-20-19600R2 

Seeing our 3D world while only viewing contour-drawings 

Dear Dr. Sawada:

I'm pleased to inform you that your manuscript has been deemed suitable for publication in PLOS ONE. Congratulations! Your manuscript is now with our production department. 

Kind regards, 

on behalf of

Dr. Markus Lappe 

Academic Editor

PLOS ONE